# Detecting Broken Strands in Transmission Lines Based on Pulsed Eddy Current

Chunhui Liao [1,*], Yinghu Yi [1], Tao Chen [1], Chen Cai [2], Zhiyang Deng [1], Xiaochun Song [1] and Cheng Lv [3]

1   Hubei Key Laboratory of Modern Manufacturing Quantity Engineering, School of Mechanical Engineering, Hubei University of Technology, Wuhan 430068, China; 101910105@hbut.edu.cn (Y.Y.); chentao@hnu.edu.cn (T.C.); dzy@hust.edu.cn (Z.D.); songxc@mail.hbut.edu.cn (X.S.)
2   Wuhan Second Ship Design and Research Institute, Wuhan 430064, China; 20160069@hbut.edu.cn
3   Hubei Special Equipment Inspection Testing Institute, Wuhan 430068, China; chenlv_nick@163.com
*   Correspondence: cloudy_c119@163.com

**Abstract:** High-voltage transmission lines are the main facilities for power transmission, and they are mainly composed of aluminum conductor steel-reinforced (ACSR). Over long-term outdoor use, overhead transmission lines will encounter lightning strikes, chemical pollutant corrosion, deicing, wind vibration, and other external forces. This often results in a series of potential failures, such as breakage, for the strands. In order to ensure the safe operation of the power grid and avoid fatal accidents, such as line breaks, it is necessary to identify and repair line faults. Among them, the main basis for the regular detection and replacement of high-voltage transmission lines is whether a broken strand defect appears. In this paper, a type of pulsed eddy current (PEC) sensor is developed to detect the broken strand defect in transmission lines. The simulation and experimental results showed that the designed PEC sensor could effectively and accurately identify the fault.

**Keywords:** transmission line; broken strands; pulsed eddy current testing

## 1. Introduction

Pulsed eddy current is a novel method for eddy current testing (ECT). The square wave excitation used in PEC can be regarded as multiple-frequency components after Fourier decomposition. PEC nondestructive testing techniques can overcome the skin effect on ECT within a certain range. Therefore, compared with single-frequency ECT techniques, PEC exhibits a greater penetration depth, and exhibits extensive applications in airplane multilayer structure detection and pipeline inspection in the petrochemical and power industries [1].

Power transmission lines are the preliminary and key facilities for power transmission, which are mainly composed of aluminum strands on the outer layer and steel strands for support. Over long-term outdoor service, overhead transmission lines may be destroyed by lightning, corrosion by chemical pollutants, ice shedding, wind-induced vibration of the conductor wires, line dance, external damages, etc. [2–5]. Some fatal accidents, such as line breakage, may be triggered if the failures cannot be recognized and repaired in time, thereby imposing serious effects on the power quality and transmission ability. In order to ensure safe operation of the power system, it is necessary to undertake periodic inspection of power transmission lines. Currently, workers mostly use telescopes to inspect potential failures along high-voltage and ultrahigh-voltage power transmission lines that require high labor intensity and low precision. Maintenance in the air with helicopters is a high-cost and fairly dangerous issue [6].

A great deal of research has been carried out on the inspection of power transmission lines, for instance, the development of robots with imaging sensors for inspection [7–9]. Using these robots, the breakage of externally stranded aluminum wires can be directly detected. However, the inspection robots are only applicable to defects on the outer surface

of power transmission lines and cannot eliminate the effects of external factors such as different illumination conditions. Based on the difference in the dissipation capability and infrared radiation between the broken and normal parts on the in-service conductors, passive infrared radiation (PIR) can be used for inspecting the break in broken lines. However, the proposed scheme is extremely sensitive to environmental temperature and may easily cause missing and false detection [10]. Several scholars propose to inspect the breakage of power transmission lines using ultrasonic waves. Using an ultrasonic transducer, an ultrasonic wave can be generated in the cable. In the case of some defects in the cable, the ultrasonic wave can be partly reflected by the transducer, i.e., the received signal can be used for defect recognition. However, an ultrasonic transducer based on piezoelectric ceramics needs an ultrasonic coupler to reduce the acoustic resistance, accompanied by poor portability [11–13]. Shoureshi et al. proposed a robot for inspecting failures in power transmission lines based on the concept of an electromagnetic acoustic transducer, and they developed a nondestructive detection system for diagnosing the mechanical integrity of the electric conductor [14]. Owing to high magnetic sensitivity, Hatsukade and Miyazaki led their teams to apply superconducting quantum interference devices (SQUIDs) with a high-temperature super (HTS) conductor to detect the breakage of a single wire in the transmission lines. Periodic distribution patterns along the breakage path could be detected in a conductor with broken lines, while no such patterns could be observed in normal conductors. These results proved the feasibility of the magnetic detection technique in detecting the breakage of a single wire. However, HTS-SQUID is too large and heavy and cannot be extensively applied in actual operating conditions [15,16]. Moreira et al. suggested that fractured steel strands can be detected with coils or permanently magnet magnetized steel strands; however, the developed sensor cannot be used for detecting the fractured Al strand in the transmission lines [17]. Yunfeng Xia et al. developed an eddy current sensor for the detection of broken Al strands in power transmission wires and the corresponding signal processing circuit. The ECT consisted of a magnetic-exciting coil and a differential detection coil. Further, the differential detection coil was composed of two identical coils via subtraction in serial; however, the ECT could only detect defects in Al wires and failed at detection in steel wires [18,19].

In order to address the existing problems in the current detection of broken strands in power transmission lines, the present study developed a pulsed eddy current sensor that simultaneously achieved the detection of Al and steel wire broken strand defects in ACSR based on the detection PEC characteristics. As is known, ACSR lines are twisted from aluminum and steel wires, and there are small gaps between the single wires. To avoid a disturbance signal resulting from these gaps, a coaxial coil is a good solution. Both driver and pick-up coils are concentric with the ACSR line as shown in Figure 1. Moreover, the number of the broken strands was also preliminarily recognized so as to provide a judgment basis for repairing the damage to the ACSR line. The detection sensor consisted of two parallel coils around the ACSR line. Specifically, one coil served as the exciting unit, while the other coil served as the signal receiving unit, which had an output voltage that was used as the signal of the PEC [20].

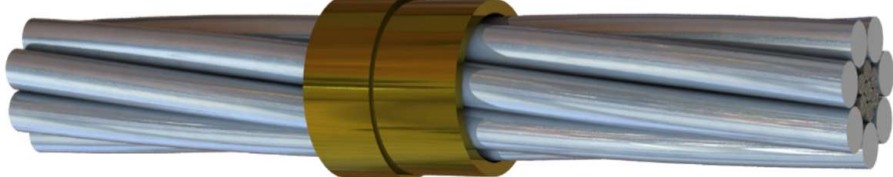

**Figure 1.** Coaxial driver coil, pickup coil, and transmission line.

## 2. Object under Investigation

In this paper, we focused on the ACSR model LGJ-120/25, and its geometric parameters are shown in Table 1. The LGJ-120/25 is widely used as a conventional detection object in transmission lines. Therefore, it is of practical engineering significance to select it as the

detection object. In the actual inspection, if the section loss of a single strand exceeds 50%, it is defined as a strand fracture. Therefore, the broken strand defect processed by us was in accordance with the inspection standard. Figure 2a shows the specimen provided by Wuhan Power Supply Bureau of Hubei Grid, China Power Grid, and Figure 2b shows the specimen with the processed broken strand defect.

**Table 1.** Parameters of the LGJ-120/25.

| | Diameter | 15.75 mm |
|---|---|---|
| ACSR LGJ-120/25 | Diameter and number of aluminum-stranded wires | 4.72 mm$^2$/7 |
| | Diameter and number of steel cores | 2.10 mm$^2$/7 |

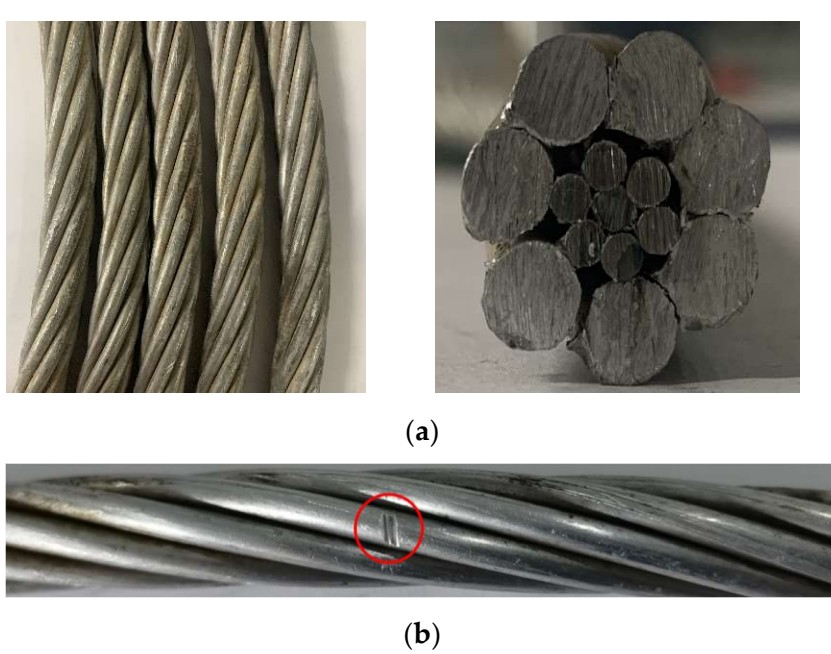

(**a**)

(**b**)

**Figure 2.** (**a**) Materials used for experiment; (**b**) broken strand defect.

## 3. Principle of Pulse Eddy Current Detection of Broken Strands in Transmission Lines with a Coaxial Encircling Coil

The pulse-excited signal is composed of a direct current (DC) component and multiple harmonic components. For ferromagnetic specimens, magnetic flux leakage is generated in the specimen under the excitation of the DC component, and eddy current signals are generated in the specimen under the excitation of various harmonic components. Therefore, PEC testing is a composite magnetic detection method by integrating magnetic flux leakage testing and ECT. Based on the characteristics of the PEC, both the external Al strand and internal steel strand of the ACSR line can be detected simultaneously. Unlike simple geometric structures, the changing spacing among strands and the lift-off change induced by the surface undulation can impose a significant effect on the output signal. The coaxial driving coil can produce a uniform eddy current along the axial direction of the specimen; thus, the influence of the lift-off change on the signal can be effectively solved by detecting the specimen around the transmission line [21].

When a pulse-excited excitation coil induces eddy current in the ACSR line through electromagnetic coupling, in turn, the circulation of the eddy current induces a secondary magnetic field. This field will vary if a flaw that impedes the eddy currents is present or there is a change in the electrical conductivity, magnetic permeability, or thickness of the sample. After receiving the magnetic field, the induction coil outputs the signal so as to analyze the condition of the ASCR line.

### 3.1. Simulation Model

COMSOL Multiphysics, as a multiphysics finite element simulation analysis software, is widely used in different simulation engineering and technology research and development. Many scholars have studied its application in electromagnetic fields [22]. Therefore, this manuscript used COMSOL Multiphysics®v. 5.5. COMSOL AB, Shanghai, China to simulate three-dimensional finite elements to analyze the pulsed eddy current magnetic field. The simulation used the infinite element domain in the setting of the physical field boundary conditions to realize the approximate simulation of the electromagnetic field, and used the impedance boundary conditions for the ACSR LGJ-120/25 specimen which reduces the calculation difficulty and realized the analysis of the characteristics of the eddy current. Physics was used to control the meshing, and the mesh size was refined. The physical field is selected as the magnetic field under the AC/DC module and researched and solved in a time-dependent manner. The three-dimensional model of PEC, in Figure 3a,b, shows the distribution of the current density obtained by simulation. Table 2 shows the parameters of the coils and specimen. In the simulation, a rectangular pulse voltage with a 50% duty cycle, 0.625 ms pulse width, amplitude of 8 V, and 15 µs rise time was used for excitation.

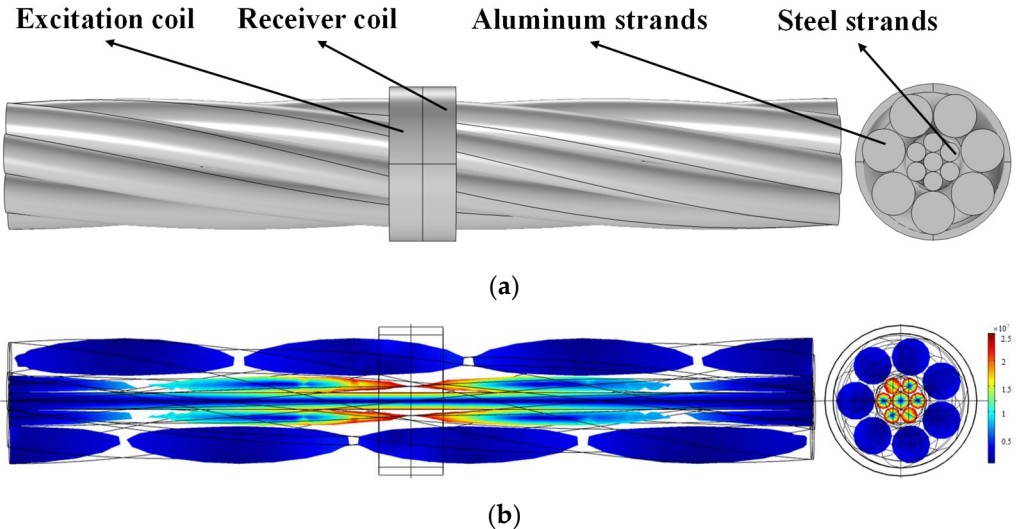

**Figure 3.** (**a**) Schematic diagram of the detection of broken strands in the transmission line based on pulsed eddy current; (**b**) the distribution of the current density during testing.

**Table 2.** Parameters used in the simulation.

| | Type | Parameter |
|---|---|---|
| | Inner diameters | 17 mm |
| | Outer diameters | 21 mm |
| PEC excitation coil | Height | 3 mm |
| | Distance between two coil axes | 0 mm |
| | Excitation coil/pick-up coil turns | 200/200 |
| ASCR LGJ-120/25 | Diameter | 15.75 mm |
| | Diameter and number of aluminum-stranded wires | 4.72 mm$^2$/7 |
| | Diameter and number of steel cores | 2.10 mm$^2$/7 |

### 3.2. Extraction of the Detected Broken Strands Features

On account of the different magnetic permeabilities, electrical conductivities, and stranded positions of the Al and steel wires, the respective broken strand defects showed

different signals in the pulsed eddy current testing. For distinguishing the defect characteristics, a simulation was performed to seek and extract the signal differences.

Figure 4a is the differential signal in the different numbers of broken Al strands. Figure 4a shows that the peak height and peak arrival time increased with the increase in the number of broken Al strands. The differential voltages of the different broken steel strand numbers are shown in Figure 4b It can be observed that the signal amplitude significantly increased with the appearance of the defects in the broken steel strands. As the number of broken strands increased, the peak value and peak arrival time increased steadily.

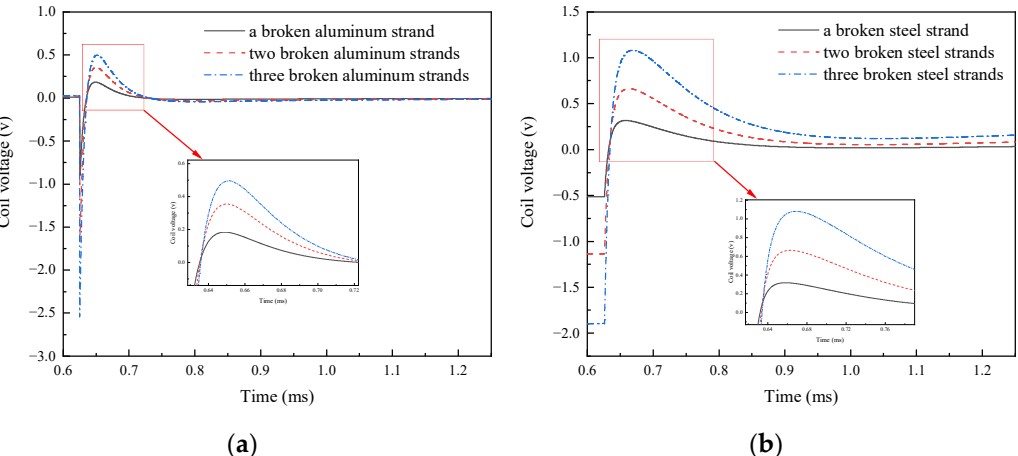

**Figure 4.** The difference signals of broken strands under different conditions: (**a**) broken aluminum strands; (**b**) broken steel strands.

The parameters commonly used in traditional defect signal analysis methods are the peak value and peak arrival time of pulse eddy current differential signals [23]. Figure 5a,b exhibit individually the relationship between the number of broken strands and the response of the pick-up coil. A linear relationship can be observed, and it indicates the number of broken strands that can be accurately detected and applied to quantitatively evaluate the defect.

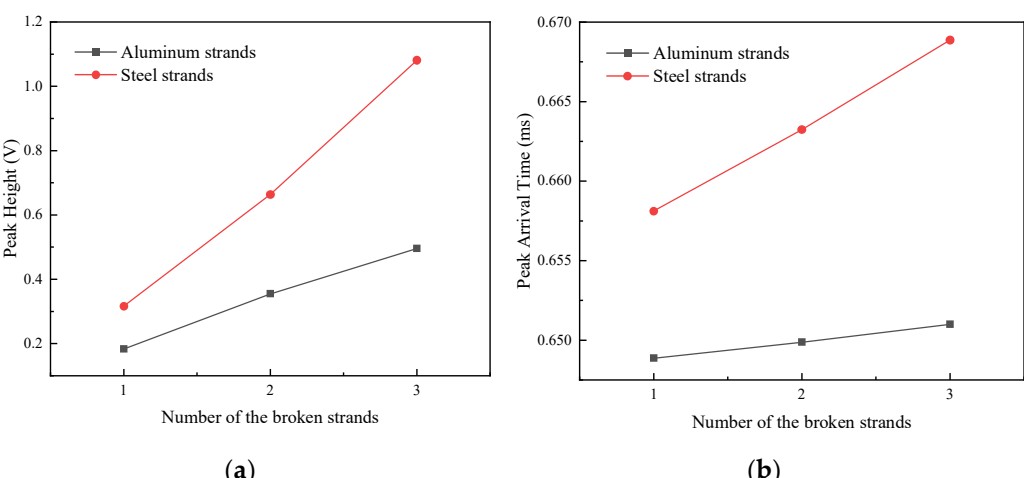

**Figure 5.** Feature peak height and peak arrival time of the different broken strands: (**a**) peak height; (**b**) peak arrival time.

In order to better classify the broken strands, the decay rate of the logarithmic scaled signal was studied, and the results are shown in Figure 6. The slope in the logarithmic scale was calculated by the equation $log(V) = p_1 t + p_2$, where the average value of $p_1$ at the end of the signal was used as the feature of the slope in the log scale [24]; $p_2$ is a constant; $t$ is the time. Figure 6a displays the case of broken Al strands, and Figure 6b shows broken steel

strands. We extracted $p_1$ to quantitative study the relationship between the defects and the response (Figure 7). From the perspective of the decay rate, the decay rate increased with the increase in the number of broken strands, and the decay rate of the steel strand was faster than the Al strand when the decay appeared in them.

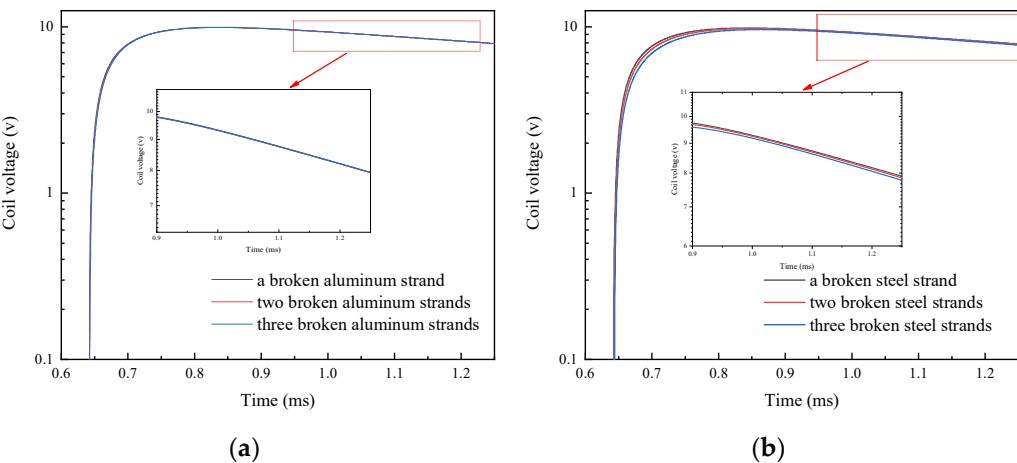

**(a)**                                                                                   **(b)**

**Figure 6.** The signals of the broken strands by the slope in the log scale under different conditions: (**a**) broken aluminum strands; (**b**) broken steel strands.

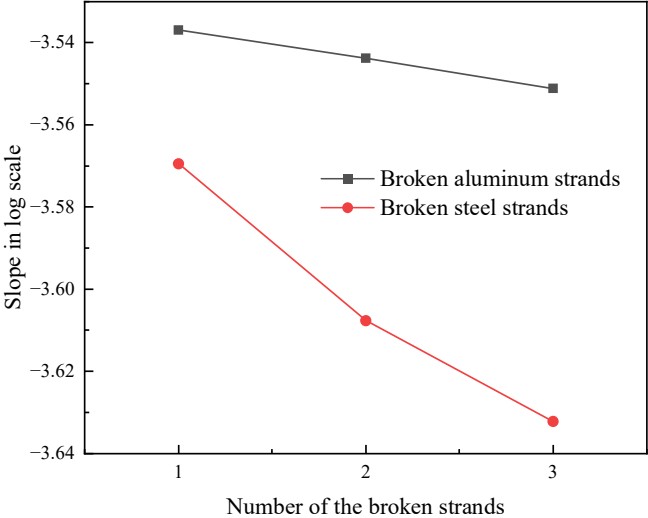

**Figure 7.** Feature of the slope in the log scale of different broken strands.

The different layers of the aluminum strand and steel strand can be approximately regarded as the problem of defect depth. It can be seen that the peak arrival time was more robust, and the peak height was also helpful for quantifying the defect size. The broken aluminum strand affected the conductivity of the sample, while the broken steel strand not only affected the conductivity but also the permeability. Because the influence of the conductivity was prominent in the rising edge, and the influence of the permeability was prominent in the stable phase of the transient response [25], the steel strand defects had a more severe impact on the later signal. Therefore, the characteristic parameters discussed in this paper can better characterize the different broken strand defects.

## 4. Experimental Validation

### 4.1. Experimental Setup

The encircling coil PEC system is shown in Figure 8, which included a signal generator, an encircling coil, a filter amplifier, an oscilloscope, a data acquisition card, and system software based on MATLAB. The signal generator was used to generate an excitation pulse

voltage with an amplitude of 8 V, frequency of 800 Hz, duty cycle of 50%, and a pulse width of 0.625 ms. The probe was composed of two coaxial circular coils. The two coils had identical parameters of 240 turns, a 23 mm outer diameter, a 17 mm inner diameter, and a 3 mm height. After a series of processing, such as filtration and amplification by the filter amplifier, and collection and output of data by the data acquisition card, the output voltage of the coil was analyzed and displayed by the specified software. A difference between the simulation and experiment was discovered in the actual experiment due to the fact that the size of the probe and material parameters were not exactly same.

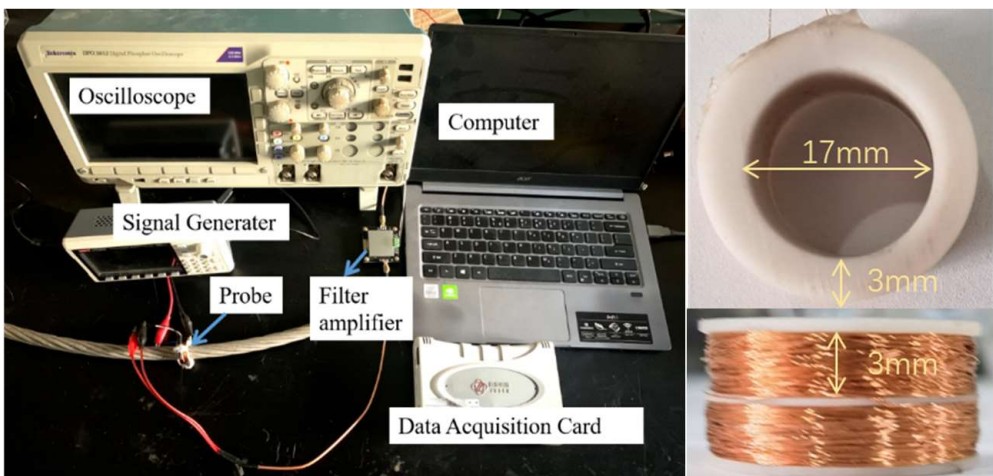

**Figure 8.** PEC experimental setup.

In the experiments, variations in the probe position on the ACSR line affected the strength of the external magnetic field, which may correspondingly affect the baseline of the PEC signal and the stability of the detection system. The baseline of the affected detection signal was recorded as $v\prime$, and the final detection signal, $v$, used for calculation was calculated by the equation $v = v\prime - m$, where $m$ is the output voltage of the sensor at the end of the pulse period [26]. The experiment was repeated 20 times on the defect-free specimen and on one broken aluminum strand. As seen from Table 3, the standard deviations of both parameters were small in the repeated experiments. This indicates that the experimental system was less affected by the background magnetic field of the probe.

**Table 3.** Repeatability of the testing results for the different probe positions.

| Experimental Parameters | Average of the Peak Value | Standard Deviation | Average Slope in Log Scale | Standard Deviation |
|---|---|---|---|---|
| Perfect transmission line | 1.6445 | 0.0028 | −2.34375 | 0.0184 |
| Broken strands in the transmission line | 1.6794 | 0.0019 | −2.42945 | 0.0090 |

*4.2. Experimental Results*

A different number of broken Al strands, where no damage to the steel strands occurred, could be detected using this method. By calculating the difference between these detection signals and the signal of complete specimen without defects, different output voltage amplitudes were obtained. The results are shown in Figure 9. As the number of broken Al strands increased, the peak value and the peak arrival time increased gradually. At the descending edge, the falling rate increased with an increasing number of broken strands. The results are consistent with Figure 4.

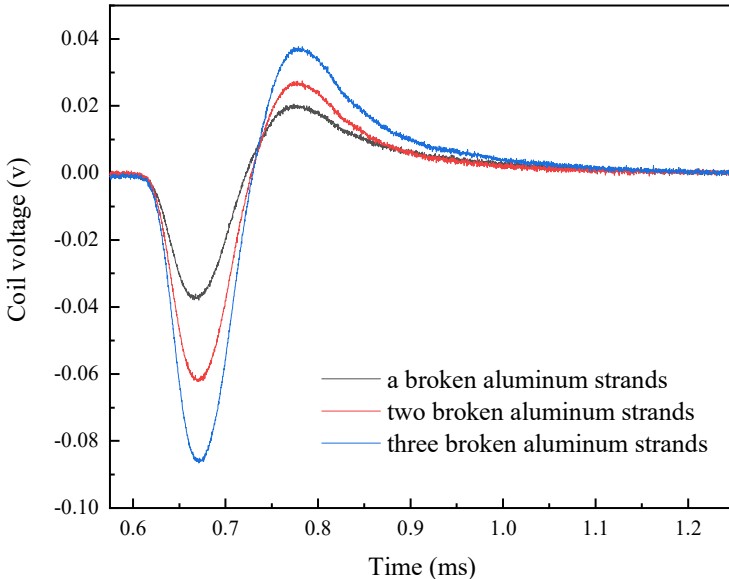

**Figure 9.** Output signals of the coil voltage with different broken aluminum strands.

Figure 10 shows the variations in the peak value and the peak arrival time with an increase in the number of broken strands. It can be observed that the peak height and peak arrival time increased with the number of broken Al strands, suggesting that the number of broken Al strands can be determined by observing the peak value and the peak arrival time.

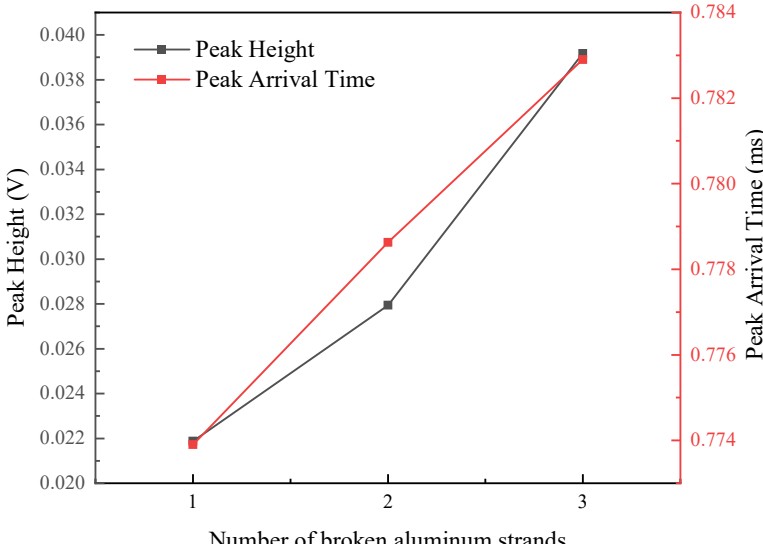

**Figure 10.** Relationship between the peak value, the peak arrival time, and the different number of broken aluminum strands.

Next, the difference between the experimental data with broken steel strands that contained no damage to the Al strands and the reference signal was calculated, and the different output voltage amplitudes were obtained (Figure 11). It can be observed from Figures 4 and 11 that the experimental data of the steel strands were consistent with the related simulation results.

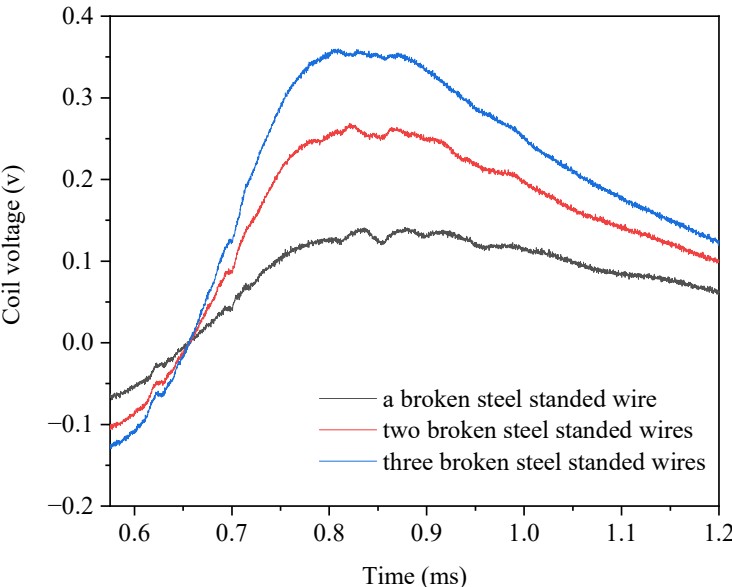

**Figure 11.** Output signals of the coil voltage with the different broken steel strands.

Figure 12 presents the variations in the peak value and the peak arrival time with an increase in the number of broken steel strands. By combining both the experimental and simulation data, it can be concluded that the peak increased steadily with an increase in the number of broken steel strands. But for the peak arrival time, the conclusion was contrary to the simulation, which may be that the irregular permeability distribution of the steel strand had an impact on the experiment.

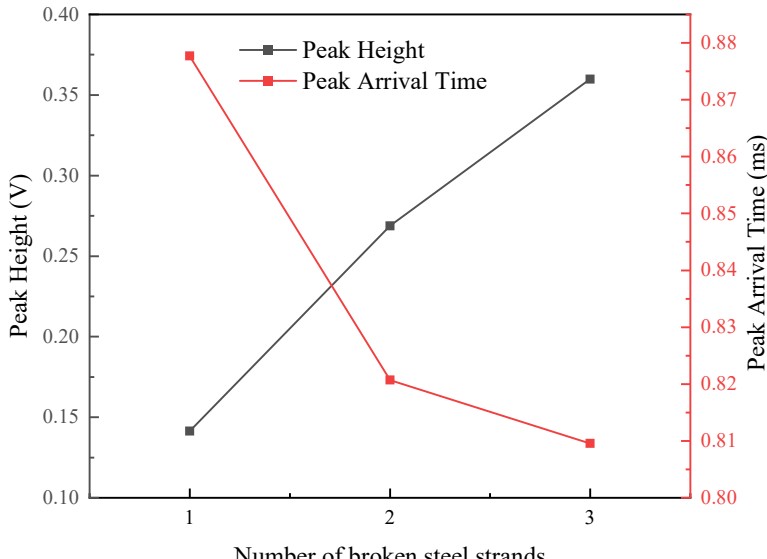

**Figure 12.** Relationship between the peak value, the peak arrival time, and the different broken steel strands.

In the experiment detecting breakages in the steel strand, the peak arrival time extracted was quite different from that of the simulation. Therefore, in the classification and identification of broken strand defects, we only considered the peak height and the slope in the log scale, and the results are shown in Figure 13. Obviously, the two classes of defects can be distinguished easily. According to defect classification criteria, a good combination should satisfy two basic criteria [27]: (1) defects of same class should go together and (2) the distance between each class is large enough. Therefore, using the characteristic parameters

of the peak height and the slope in the log scale, it can be successfully judged whether the broken strand defect occurred in the steel strand or aluminum strand.

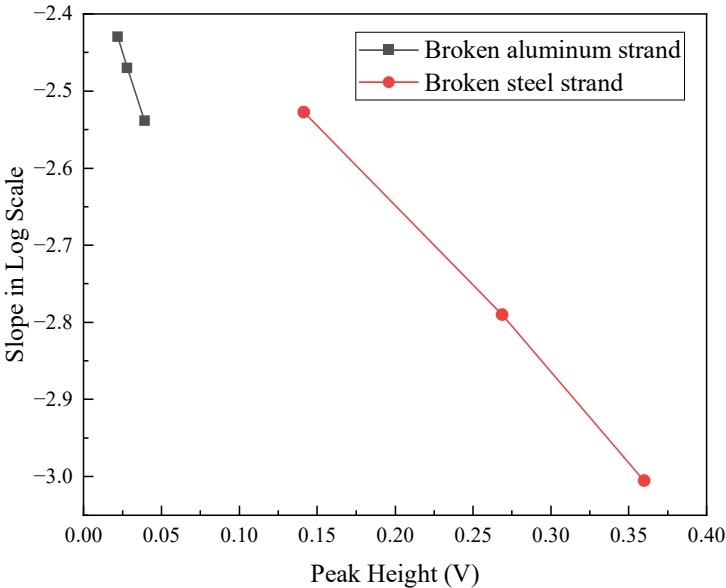

**Figure 13.** Defect classification recognition using peak height and slop in the log scale.

The present study confirmed the feasibility of the slope in the logarithmic scale, peak arrival time, and the peak value of the PEC signal in characterizing the broken strand defect in the ACSR line. The broken strands in both the steel and Al strands could be simultaneously detected by observing the variations in the characteristic parameters (namely, the slope in the logarithmic scale, peak arrival time, and the peak value) in the same detection signal or considering the derivation based on these characteristic parameters.

## 5. Conclusions

This study investigated a defect detection method for broken strands in ACSR lines based on coaxial coils under the excitation of the pulsed eddy current. Accordingly, the broken strands in the outer-layer Al strands and inner-layer steel strands of the power transmission lines were simultaneously detected. It provides a new reference method for detecting broken strands in transmission lines. The important conclusions are described as follows:

1.  From the modeling simulation and experimental results on Al strands with a different number of broken strands, it was found that the broken strands in the Al strands could change the coil voltage and lead to the appearance of the zero-crossing phenomenon. As the number of broken strands increased, both the decay rate, peak arrival time, and the peak value of coil voltage increased;
2.  The broken strands in the steel strands could induce an increase in the voltage of the detection coil. The change in the coil voltage amplitude was positively correlated with the number of broken strands. In addition, the coil voltage dropped gradually after reaching a peak value, and the decay rate increased with the increase in the number of broken strands;
3.  Based on the difference in the detection signals between the broken strands in the steel and Al strands, the slope in the logarithmic scale and the peak value were determined as the characteristic parameters in the detection of broken strands in ACSR lines. Moreover, the feasibility of the proposed detection method was validated via experiment and simulation. Thus, the proposed method can provide an effective means for locating and quantitatively recognizing broken strands in power transmission lines.

## 6. Future Developments

Although the designed method performed as expected, there are still some shortcomings such as installation of in-service inspection. Therefore, additional developments are required before it is suitable for use in a commercial environment.

In the laboratory, we had a simple cut of cable so we could put the circular excitation and detection coils from any side. However, practically, cables are continuous between the poles; if in-service tests are required, an open design for the coil is preferred. A flexible flat cable (FFC) or a flexible printed circuit board (FPCB) may be a good choice, and the coil can be easily installed and serviced.

In a real device, storing and subtracting a sample signal is probably not practical. It is necessary to use a differential probe to optimize the probe, which is conducive to improving the quality of the signal. In addition, the magnitude B field induced by the current in transmission lines will affect detection; this issue will be further investigated in follow-up research.

**Author Contributions:** C.L. (Chunhui Liao) proposed the idea and helped in writing and revision of the manuscript; Y.Y. contributed to the theory research, the design of experiments, writing and revision of the manuscript; T.C., C.C., Z.D., X.S. and C.L. (Cheng Lv) contributed to the design of the experiments and helped in revision of the manuscript. All authors have read and agreed to the published version of the manuscript.

**Funding:** This research was funded in part by the National Natural Science Foundation of China (grant number: 51807052), the Research on Key Technologies of Magnetized Array Eddy Current Wall Climbing Robot (grant number: Hbscjg-JS2021006), and the Youth Project National Natural Science Foundation of China (grant number: 52105550).

**Institutional Review Board Statement:** Not applicable.

**Data Availability Statement:** Not applicable.

**Acknowledgments:** Materials used for the experiments in this paper were donated by the Wuhan Power Supply Bureau of Hubei Grid, China Power Grid.

**Conflicts of Interest:** The authors declare no conflict of interest.

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
