# Peer review of "Detecting Broken Strands in Transmission Lines Based on Pulsed Eddy Current"

_metals, doi:10.3390/met12061014_

Round 1
Reviewer 1 Report
The submitted version is not a final one! Part of text is in red, figure numbers and citation numbers are missing. Reviewing such a paper never happened to me.
1) line 62, text "pattern s along"
2) comment to text around figure 1: what is clearance between coil and power transmission line.
How you continue survey when inspection goes over the transmission line pole.
3) line 114, comment: Please explain in few words what it means as it is essential for understanding the concept. I know that it is explained in [21], but for sake of simplicity I suggest to add short explanation.
4) Figure 3: Is image 3b), left part cross section or outer view as the 3a) left part?
5) starting from line 142 the text is in red letters!!!
6) line 170. There is a figure in lnie 170 without figure number and the text " the response ( Figure)."!!!
7) Figure 7: There is no referring to figure 7 in text. You are not mentioning or commenting figure 7!
8) Figure 8: Show the detail image of the probe (coil)!!!
It is essential.
9) Line 209 "The results are shown in Error! Reference source not 209 found., as the number of broken"
10) Line 215:"Figure shows", missing figure number
11) Lines 225 and 226:"obtained (Figure ). It can be observed from Figure 4 and Figure that the experimental"
12) Line 230: missing figure number!
13) Figure 13: not clear. Are all blue and red lines broken ones or only thick red and blue lines?
Figure 13 has to be explained more deeply!
Reviewer 2 Report
I was really looking forward to reading this paper. The problem it seeks to address is certainly an important one. We have done some work on this and, also, are big fans of pulsed eddy current. I think the authors are a bit naive in their approach but I enjoyed the paper. I really have to question what it is doing in this journal. I would have thought that something like Journal of Nondestructive Evaluation would be a much more appropriate venue. If the authors want to stay with MDPI, then perhaps Sensors would be a better venue. Metals seems to me to be a materials based journal and this is really all about non-destructive evaluation.
I have listed a number of issues with the paper below. For the most part these should be easy to deal with. I think the authors need to give a lot more consideration to how a practical system might work. For instance, how would they get their probe onto a transmission line? How fast can it go? If we said one coil height per cycle then we are talking 800 /s x 3 mm = 2.4m/s which is probably at the bottom end of what would be acceptable.
Why is some of the document in red?
Section 3.1 Simulation – need to know the number of turns in the drive and pick-up coils and the gauge of the wire. Also, there should be some discussion of boundary conditions. Basically, I should have enough info to repeat your calculation (assuming I have COMSOL)
Presumably, you have done something to prevent wire to wire conduction which would let current bypass a broken wire. Possibly shrunk your wires slightly so they don’t touch.
Table 2 values for ID and OD of the coil are reversed.
Given your experimental results, I wonder (again) what you used for # of turns in your drive coil in the simulation (as mentioned above). A 15 us rise time corresponds to a 70 kHz frequency and the inductance of your drive coil could become a significant problem.
Section 3.2
It would be useful to know what the signals looked like without the reference subtracted. Getting a valid reference for an actual application can be tricky.
Clearly we are only using the falling edge here. I don’t see why we couldn’t use the rising edge as it should e the same just inverted. That would let you go twice as fast.
Speaking of that, there is something wrong with fig 4b. The signal coming in from 0.6 ms should be at the negative of the signal going out at 1.25 ms and clearly, it isn’t.
Line 156 should read “the features commonly used” These aren’t eigenvalues of anything.
line 163-179. This is using the wrong part of the signal. Obviously, the steel signal decays more slowly than the aluminum but only in the time range 0.625- 0.9 ms. I seriously doubt that that difference in slope could be picked up in a real life application on a real transmission line. Rather than using the zero crossing time you have used, which is so short it probably can’t be reliably measured in a real system why not pick something like the time the signal is above 0.1V. That should allow you to easily distinguish between steel and aluminum and then you can use peak height for the number of broken strands.
Section 4.1
What gauge of wire? Did you measure the inductance of your drive coil and its resistance?
Line 195 I don’t understand the comment about background field. Are they referring to the field being produced by the cable – which could be very substantial.
Line 196 vused should be v(nu) used.
I think this is a bit niave. In a real device storing and subtracting a sample signal is probably not practical. Possibly one could have a second pick-up at the other end of the drive that could serve as a reference under the assumption the defect was very localized. Possibly you would want a second drive and pick up that would trail the primary by a few cm. The quality of the reference signal will always be a problem because the coils will bounce around the cable a bit with the necessary clearance. There will be a fine balance between minimizing the clearance and not binding on the cable.
Another issue here is that the device should be moving at speed along the line. It would be helpful if, in the simulations, the authors showed how the signal depends on motion of the system.
Line 200 was the probe moved between measurements. That could increase the standard deviation significantly.
Line 209 reference to fig 9 has been lost.
Line 215 Figure ?
Line 230 Figure?
Fig 13. Get rid of the projections on the walls. They only confuse things.
Round 2
Reviewer 1 Report
No other comments.
Author Response
Special thanks again to you for your observation and suggestions.
Reviewer 2 Report
So for the most part the authors have dealt with my initial comments. I disagree with some of their opinions in their reply, but those shouldn’t hold up publication. In a couple of places, it is clear that we had a miscommunication. For instance, on point 3, my comment on boundary conditions was just a question of whether they used the default COMOSL boundary condition “magnetic insulation” as opposed to something like a periodic condition on the ends of the wire. Personally, I don’t think we need the added lines 131-154. If you are using COMSOL, you should know this info.
Point 4. I assume figures 4 and 5 come from the simulation. I further assumed that when the authors say “broken” they meant 100%. If they mean a 50% reduction then they should clearly say so. I would also add a line about what “broken” means to the introduction. If the line is 100% broken and the lines in the simulation are touching, then we can get current transfer between the lines. In real life, such transfer would have to overcome the surface resistance between wires, which would be large relative to the conduction of the wires. If this is ignored you can get unrealistic current flow. In some of the simulations I have seen, the authors shrank the wires slightly to avoid touching (which also causes meshing issues). In any case, the authors should indicate how they modelled the “breaks”. If the wires aren’t 100% broken then clearly this will result in a hot spot in the wire and a different resistivity. I don’t expect the authors to really deal with this issue, but they should at least mention it. It would be a nice demonstration of the “multiPhysics” part of COMSOL. I also suspect that the authors did not model any large AC currents flowing in the wires. Somewhere in the discussion section they should probably mention how they would expect this to affect their measurements. (e.g. Need for a high pass filter on the pick-up, slow change relative to pulse signal, etc.).
The only other point I have is that they missed my point (10) which is now on line 181. Peak value and arrival time are not eigenvalues, they are parameters. Eigenvalues have to do with linear transformations where a vector/function is transformed into itself (eigenvector or eigenfunction) times a constant (eigenvalue).
Author Response
Please see the attachment

This manuscript is a resubmission of an earlier submission. The following is a list of the peer review reports and author responses from that submission.
Round 1
Reviewer 1 Report
The manuscript is about the Detecting broken strands in transmission line using pulsed Eddy Current testing, the manuscript address an interesting point of research. It is interesting and well presented. However it does need English correction (for example in line 22 ‘’where in the excited exciting square wave’’) and many typographical mistakes (for example in line 24 technique s can overcome) in the manuscript.
The pulsed eddy current is not very new, however it is well adapted by the researches for the corrosion detection and testing of multi-layer structures.
It would be better if authors specify the novelty of the method. Moreover the authors use regular time domain features like time to peak and decay time etc.
As it is seen that the detection coil is circular, it is wound around the testing cable (how can we put the cable in to the centre of the probe we have to cut the cable and put it through the centre of the coil of we have to wind the excitation coil and detection coil around the cable?), how it can be fixed around the power cable? Is it practically possible? Because in the laboratory we have a simple cut of cable so we can put the circular excitation and detection coils from any side. But practically the cable are continuous between the poles. Please discuss the practicality of the method.
It is mentioned that the signal is differential but it looks the probe has single detection coil, please mention how the differential signal is obtained.
In the fig.3 simulation results, the eddy current density is seen only in the steel wire, but not in the external Al wire. Is it because of skin effect? It would be better to specify the skin depth of AL and Steel at the specified pulse width and repetition frequency.
Please check the Fig.4 and 5, figure captions and figure legends, for example in Fig.4 says (a) steel stranded wire integrity; (b) a broken steel stranded wire, while the figures shows the broken aluminium strands. Authors should check these things carefully.
The induced current density may depends on the imposed magnetic field strength, and more over it has to go deeper steel sires below the AL strands, please mention what is the current of the excitation pulse.
Reviewer 2 Report
- Some surprise for the reader is the appearance of Fig. 4, Fig. 5 and Fig. 6. The authors write about some simulation. What was this simulation about? What calculation tools were used? What equations or formulas were used? Authors should supplement the article with answers to the above questions.
- Next the authors write about combining the theoretical and simulation analysis. What was that theoretical analysis? This should, at least briefly, be presented in the paper.
- The fundamental question is the practical application of the proposed method in the case of a power transmission line during its operation. After all, examining failures from a distance with the proposed method is impossible. How then to install these two coils on the wires? I would expect some suggestions from the authors on this matter.
Reviewer 3 Report
The authors have addressed a general issue in industrial environments. The manuscript describes eddy current inspection of power transmission lines consisting of two different material strands. However, the manuscript lacks integrity and soundness in many different aspects. My comments are as follows:
General comments:
- English writing is very poor, and, in many cases, wrong grammar has been used. Therefore, the whole manuscript needs to be rewritten and proofread.
- The quality of images and graphs is low and not consistent. For example, Figure 3(b): scale/legend is not readable at all. Other images are low quality.
- Mind using acronyms in the proper way. Define the acronym in the first use and use it throughout the manuscript.
Specific comments:
- Introduction, lines 22 to 24: What do mean 'Pulsed eddy current (PEC) is a novel method for detecting the eddy current' and 'excited exciting square wave'? non-destructive detection to be replaced by non-destructive testing. In the same paragraph, line 26, 'eddy current detection' has to be replaced with 'eddy current testing'.
- Introduction, line 38: what is the significance of 'Telescope'?
- Introduction, line 47: what is the significance of 'real-time environment'?
- Introduction, line 47: passive infrared thermography is a scheme of infrared thermography and is not a sensor. The same applies to line 51.
- Introduction, lines 75, 76: 'ECT can only detect the defects in Al wires and fails in the detection of steel wires', how did you come to this conclusion? As far as I read these two references, they haven't mentioned this capability.
- Introduction, lines 79, 80: 'achieved the detection of defects in steel-cored Al strands and broken strands in steel wires based on the detection PEC characteristics, the structure is shown in Figure1' is not clear what you mean.
- In your model you have considered off-line testing, meaning when there’s no current in the transmission line. However, the real case implies the flowing current and the resulting magnetic field to be considered as per references 18 and 19.
- Section 2, line 89: please define 'ASCR model LGJ-120 / 25'.
- Section 2, Table 1: Diameter is mentioned in mm2 (this also applies to Table 2). Please define either the diameter or the cross-section area.
- Section 3, line 118: 'When the pulse-excited current poses excitation on the coil,' please rephrase and use the correct structure.
- Section 3, lines 120-122: Please rethink the eddy current generation, skin effect and Joule effect. The skin effect is not equivalent to the Joule effect and vice-versa. Also, in lines 123 and 14, please explain better the eddy current generation and its relationship with primary and secondary fields.
- Section 3, lines 126-129: 'For ferromagnetic specimen, the corresponding ....' please review and rewrite the sentence in the correct and understandable way.
- Section 3, line 131, 132: 'Based on the characteristics of the pulsed eddy current ....' the sentence is incomplete.
- Section 3.1, lines 136, 137: 'current density magnetic flux density distribution obtained by simulation', Figure shows which of these two parameters?
- Section 3.1, lines 137, 138: too many times using the word 'parameter'. please rephrase.
- Section 3.1, lines 139, 140: what is the amplitude of the pulse voltage?
- Section 3.2, line 149: There's no 'magnetic conductivity' but 'magnetic permeability'.
- Section 3.2, lines 149-151: ' ... and stranded position between the Al strand and steel strand in the steel-cored Al strand show different signals in pulsed eddy current detection.' I failed to understand what you mean by this sentence!
- Section 3.2, lines 158-160: What is the significance of zero-crossing in the signals and how do you interpret it in this case?
- Figure 4(a,b):
- Why does the time axis start from 2.4 ms?
- Are you sure the voltage scale on the y-axis is in V for the differential signal?
- Why do the signals in the graph (b) start always from -0.2 V and never decay toward zero but +0.2 V?
- Figure 5(a,b):
- Why does the time axis start from 2.4 ms?
- Are you sure the voltage scale on the y-axis is in V for the differential signal?
- Why do the signals in both graphs start from different values ranging from 0.25 V to almost 0.75 V in almost equal steps and never decay toward zero?
- Figure 6(a,b): No need for 1.5 and 2.5 on the x-axis, moreover, considering figures 4 and 5, you have only examined two cases; one broken steel/aluminium strand or all intact strands and for each case, you have investigated 1 to3 broken strands in the other material. This doesn’t match with the results presented in Figure 6.
- Speaking about the slope and decay rate, I would strongly recommend showing the graphs of the signals on the log-log scale and then showing the slope by fitting a line on the curves to show the correctness of your process.
- Page 7, lines 194 to 217: I would suggest combining both paragraphs and explaining the reasons you have mentioned in the second paragraph for each observation in the first paragraph.
- Page 7, lines 201-203: please better explain this ‘From the perspective of the decay rate, the decay rate of Al wire is faster than steel core when there the decay appears in them.’.
- Lines 208 and 211: please refer to comment #17.
- Section 4.1, line 223: How much is the pulse width?
- Section 4.1, line 227: ‘the put voltage’ probably output.
- Page 8, line 235: ‘baseline was affected was recorded’ was recorded or affected?
- Page 238, line 238: ‘1 broken strand of aluminium strand’ please correct it.
- A general comment about the design of the experiments and simulation: You have considered the coils to be two concentric rings of the same size encircling the wire, my question is in the real world how you can make such a coil and insert the wire into it in the field? This looks more hypothetical than the real pragmatic solution. It may work in the laboratory when you have a piece of wire but not in real applications.
- Figure 8: Why does the time axis start from 0.6 ms? Comparing Figure 8 and Figure 4(a), they look quite different in terms of time scale and amplitude scale. If we consider the same signal and the same sample, they are far different, how do you explain this?
- Figure 9: No axis label for the right y-axis values. No necessity to put 1.5 and 2.5 ticks.
- Figure 10: please refer to comment 32.
- Figure 11: Please refer to comment 33.
- I strongly recommend combining Figures 11 and 9 to show how you can distinguish and separate the defect in steel strands from aluminium strands by the slope and peak value and peak time.
- I recommend improving Section 5 (Conclusions) with more discussions on the results and considering the comments mentioned before in order to demonstrate concrete reasoning for your research work and findings.
References: I suggest improving the bibliography by referring to more relevant works.
Considering all comments above, I would like to ask the authors to do a major revision of their manuscript considering a sound and precise theoretical and experimental base and resubmit their manuscript after such improvement, since the manuscript in its current form is not suitable for publication.
Best of luck!
Round 2
Reviewer 3 Report
Thanks to the authors for their reply to the comments. However, I am not unfortunately satisfied with their answers, which in my opinion is because they didn't well dig into the concept of eddy current inspection. The biggest drawback of the manuscript is that simulation is not supporting the experiments and vice versa. The assumptions are very simple and underestimate many important parameters.
Unfortunately, I cannot accept the manuscript in its current form.